# LEARNING VECTOR REPRESENTATION OF LOCAL CONTENT AND MATRIX REPRESENTATION OF LOCAL MOTION, WITH IMPLICATIONS FOR V1

## ABSTRACT

This paper proposes a representational model for image pair such as consecutive video frames that are related by local pixel displacements, in the hope that the model may shed light on motion perception in primary visual cortex (V1). The model couples the following two components. (1) The vector representations of local contents of images. (2) The matrix representations of local pixel displacements caused by the relative motions between the agent and the objects in the 3D scene. When the image frame undergoes changes due to local pixel displacements, the vectors are multiplied by the matrices that represent the local displacements. Our experiments show that our model can learn to infer local motions. Moreover, the model can learn Gabor-like filter pairs of quadrature phases.

## 1 INTRODUCTION

Our understanding of the primary visual cortex or V1 (Hubel & Wiesel, 1959) is still very limited (Olshausen & Field, 2005). In particular, the mathematical and representational models for V1 are still in short supply. Two prominent examples of such models are sparse coding (Olshausen & Field, 1997) and independent component analysis (ICA) (Bell & Sejnowski, 1997). Although such models do not provide detailed explanations of V1 at the level of neuronal dynamics, they help us understand the computational problems being solved by V1.

In this article, we propose a model of this sort. It is a representational model of natural image pair that are related by local pixel displacements. The image pair can be consecutive frames of a video sequence, where the local pixel displacements are caused by the relative motions between the agent and the objects in the 3D environment. Perceiving such local motions can be crucial for inferring ego-motion, object motions, and 3D depth information.

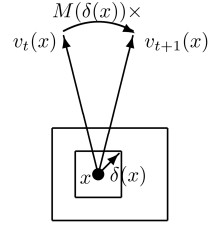

As is the case with existing models, we expect our model to explain only limited aspects of V1, some of which are: (1) The receptive fields of V1 simple cells resemble Gabor filters (Daugman, 1985). (2) Adjacent simple cells have quadrature phase relationship (Pollen & Ronner, 1981). (3) The V1 cells are capable of perceiving local motions. While existing models can all explain (1), our model can also account for (2) and (3) naturally. Compared to models such as sparse coding and ICA, our model serves a more direct purpose of perceiving local motions.

Figure 1: Scheme of representation

Our model consists of the following two components. See Figure 1 for an illustration, where the image is illustrated by the big rectangle. A pixel is illustrated by a dot. The local image content is illustrated by a small square around it. The displacement of the pixel is illustrated by a short arrow, which is within the small square. The vector representation of the local image content is represented by a long vector, which rotates as the image undergoes deformation due to the pixel displacements. Section 3 explains the notation.

(1) Vector representation of local image content. The local content around each pixel is represented by a high dimensional vector. Each unit in the vector is obtained by a linear filter. These local filters

or wavelets are assumed to form a normalized tight frame, i.e., the image can be reconstructed from the vectors using the linear filters as the basis functions.

(2) Matrix representation of local displacement. The change of the image from the current time frame to the next time frame is caused by the displacements of the pixels. Each possible displacement is represented by a matrix that acts on the vector. When the image changes according to the displacements, the vector at each pixel is multiplied by the matrix that represents the local displacement, in other words, the vector at each pixel is rotated by the matrix representation of the displacement of this pixel.

One motivation of our work comes from Fourier analysis. An image patch $\mathbf{I}$ can be expressed by the Fourier decomposition $\mathbf{I} = \sum_k c_k e^{i\langle \omega_k, x \rangle}$. Assuming the image patch undergoes a smooth motion so that all the pixels are shifted by a constant displacement $dx$, the shifted image patch $\mathbf{J}(x) = \mathbf{I}(x - dx) = \sum_k c_k e^{-i\langle \omega_k, dx \rangle} e^{i\langle \omega_k, x \rangle}$. The change from the complex number $c_k$ to $c_k e^{-i\langle \omega_k, dx \rangle}$ corresponds to rotating a 2D vector by a $2 \times 2$ matrix. However, we emphasize that our model does not assume Fourier basis or its localized version such as Gabor filters. The model figures it out with generic vector and matrix representations.

We train this representational model on image pairs where in each pair, the second image is a deformed version of the first image, and the deformation is known. We learn the encoding matrices for vector representation and the matrices that represent the pixel displacements from the training data.

Our experiments show that our method can learn V1-like units that can be well approximated by Gabor filters with quadrature phase relationship. After learning the encoding matrices for vector representation and the matrix representations of the displacements, we can infer the displacement field using the learned model. Compared to current optical flow estimation methods (Dosovitskiy et al., 2015; Ilg et al., 2017), which use complex deep neural networks to predict the optical flow, our model is much simpler and is based on explicit vector and matrix representations. We also demonstrate comparable results to these methods, in terms of the inference of displacement field.

In terms of biological interpretation, the vectors can be interpreted as activities of groups of neurons, and the matrices can be interpreted as synaptic connections. See subsections 4.3 and 4.4 for details.

## 2 CONTRIBUTIONS AND RELATED WORK

This paper proposes a simple representational model that couples the vector representations of local image contents and matrix representations of local pixel displacements. The model is new and different from existing models for V1. It explains some aspects of V1 simple cells such as Gabor-like receptive fields and quadrature phase relationship. It adds to our understanding of V1 motion perception in terms of a representational and relational model.

The following are two themes of related work.

(1) V1 models. Most well known models for V1 are concerned with statistical properties of natural images or video sequences. Examples include sparse coding model (Olshausen & Field, 1997; Lewicki & Olshausen, 1999; Olshausen, 2003), independent component analysis (ICA) (Hyvärinen et al., 2004; Bell & Sejnowski, 1997; van Hateren & Ruderman, 1998), slowness criterion (Hyvärinen et al., 2003; Wiskott & Sejnowski, 2002), and prediction (Singer et al., 2018). While these models are very compelling, they do not serve a direct purpose of perceptual inference. Our model is learned for the direct purpose of perceiving local motions caused by relative motion between the agent and the surrounding 3D environment.

We want to emphasize that our model is complementary to the existing models for V1. Similar to existing models, our work assumes a linear generative model for image frames, but our model adds a relational component with matrix representation that relates the consecutive image frames. Our model is also complementary to slowness criterion in that when the vectors are rotated by matrices, the norms of the vectors may remain constant.

(2) Matrix representation. In representation learning, it is a common practice to encode the signals or states as vectors. However, it is a much less explored theme to represent the motions, actions or relations by matrices that act on the vectors. An early work in this theme is (Paccanaro & Hinton, 2001), which learns matrices to represent relations. More recently, (Jayaraman & Grauman, 2015)

learns matrix representation for ego-motion. (Gao et al., 2018) learns vector representation for self-position and matrix representation for self-motion in a representational model for grid cells. Our work constitutes a new development along this theme.

The matrix representation of local displacements in our work is partially inspired by the group representation theory, where the group elements are represented by matrices acting on the vectors (Fulton & Harris, 2013). In our work, local displacements belong to 2D Euclidean group. Our modeling of local motion dx is similar to the treatment of Lie group via Lie algebra by analyzing infinitesimal changes. The objects in the image may undergo more complex motions which form more complex Lie groups (e.g., rotations and translations). We can again represent the objects (e.g., their poses) in the scene by vectors, and represent the motions of the objects by matrices. The representation theory underlies much of modern mathematics and holds the key to the quantum theory (Zee, 2016). Perhaps it also underlies the visual and motor cortex, where the neuron activities encode vectors, and the synaptic connections encode the matrices that rotate them, with the matrices representing motions, actions, and relations.

## 3 REPRESENTATIONAL MODEL

### 3.1 VECTOR REPRESENTATION

Let $\{\mathbf{I}(x), x \in D\}$ be an image observed at a certain instant, where $x = (x_1, x_2) \in D$ is the 2D coordinates of pixel. $D$ is the image domain (e.g., $128 \times 128$). We represent the image $\mathbf{I}$ by vectors $\{v(x), x \in D_-\}$, where each $v(x)$ is a vector defined at pixel $x$, and $D_-$ may consist of a sub-sampled set of pixels in $D$ (e.g., sub-sampled every 8 pixels). $V = \{v(x), x \in D_-\}$ forms a vector representation of the whole image.

We assume the vector encoding is linear and convolutional. Specifically, let $\mathbf{I}[x]$ be a squared patch (e.g., $16 \times 16$) of $\mathbf{I}$ centered at $x$. We can make $\mathbf{I}[x]$ into a vector (e.g., 256 dimensional). Let

$$v(x) = W\mathbf{I}[x], \ x \in D_-, \tag{1}$$

be the linear encoder, where $W$ is the encoding matrix that encodes $\mathbf{I}[x]$ into a vector $v(x)$, and $W$ is the same for all $x$, i.e., convolutional. The rows of $W$ are the linear filters and can be displayed as local image patches of the same size as the image patch $\mathbf{I}[x]$. We can write $V = \mathbf{W}\mathbf{I}$, if we treat $\mathbf{I}$ as a vector, and the rows of $\mathbf{W}$ are the shifted or translated versions of $W$.

### 3.2 NORMALIZED TIGHT FRAME AND ISOMETRY

We assume that $\mathbf{W}$ is an auto-encoding normalized tight frame, i.e.,

$$\mathbf{I} = \mathbf{W}^\top V, \tag{2}$$

Thus, the linear filters for bottom-up encoding also serve as basis functions for top-down decoding. Both the encoder and decoder can be implemented by convolutional linear neural networks.

The normalized tight frame assumption can be justified by the fact that for two images $\mathbf{I}$ and $\mathbf{J}$, we have $\langle \mathbf{W}\mathbf{I}, \mathbf{W}\mathbf{J} \rangle = \mathbf{I}^\top \mathbf{W}^\top \mathbf{W}\mathbf{J} = \langle \mathbf{I}, \mathbf{J} \rangle$, that is, the vector representation preserves the inner product. As a result, $\|\mathbf{W}\mathbf{I}\| = \|\mathbf{I}\|$, $\|\mathbf{W}\mathbf{J}\| = \|\mathbf{J}\|$, thus the vector representation also preserves the angle and has the isometry property.

When the image $\mathbf{I}$ changes from $\mathbf{I}_t$ to $\mathbf{I}_{t+1}$, its vector representation $V$ changes from $V_t$ to $V_{t+1}$, and the angle between $\mathbf{I}_t$ and $\mathbf{I}_{t+1}$ is the same as the angle between $V_t$ and $V_{t+1}$.

### 3.3 SUB-VECTORS

The vector $v(x)$ can be high-dimensional. We further divide $v(x)$ into $K$ sub-vectors, $v(x) = (v^{(k)}(x), k = 1, ..., K)$. Each sub-vector is obtained by an encoding sub-matrix $W^{(k)}$, i.e., $v^{(k)}(x) = W^{(k)}\mathbf{I}[x], \ k = 1, ..., K$, where $W^{(k)}$ consists of the rows of $W$ that correspond to $v^{(k)}$. According to the normalized tight frame assumption, we have $\mathbf{I} = \sum_{x \in D_-} \sum_{k=1}^{K} W^{(k)\top} v^{(k)}(x)$. In practice, we find that this assumption is necessary for the emergence of V1-like receptive field.

### 3.4 MATRIX REPRESENTATION

Let $\mathbf{I}_t$ be the image at time frame $t$. Suppose the pixels of $\mathbf{I}_t$ undergo local displacements, where the displacement at pixel $x$ is $\delta(x)$. We assume that $\delta(x)$ is within a squared range $\Delta$ (e.g., $[-6, 6] \times [-6, 6]$ pixels) that is inside the range of $\mathbf{I}_t[x]$ (e.g., $16 \times 16$ pixels). Let $\mathbf{I}_{t+1}$ be the resulting image. Let $v_t(x)$ be the vector representation of $\mathbf{I}_t[x]$, and let $v_{t+1}(x)$ be the vector representation of $\mathbf{I}_{t+1}[x]$. Then $v_t(x) = (v_t^{(k)}(x), k = 1, ..., K)$, and $v_{t+1}(x) = (v_{t+1}^{(k)}(x), k = 1, ..., K)$.

The transition from $\mathbf{I}_t$ to $\mathbf{I}_{t+1}$ is illustrated by the following diagram:

$$
\begin{array}{ccc}
v_t^{(k)}(x) & \xrightarrow{\;M^{(k)}(\delta(x))\times\;} & v_{t+1}^{(k)}(x) \\[4pt]
W^{(k)} \uparrow & \underset{\delta(x)}{\uparrow} & \uparrow W^{(k)} \\[4pt]
\mathbf{I}_t & \xrightarrow{\hspace{2cm}} & \mathbf{I}_{t+1}
\end{array}
\tag{3}
$$

Specifically, we assume that

$$v_{t+1}^{(k)}(x) = M^{(k)}(\delta(x))v_t^{(k)}(x), \; \forall x \in D_-, k = 1, ..., K. \tag{4}$$

That is, when $\mathbf{I}$ changes from $\mathbf{I}_t$ to $\mathbf{I}_{t+1}$, $v^{(k)}(x)$ undergoes a linear transformation, driven by a matrix $M^{(k)}(\delta(x))$, which depends on the local displacement $\delta(x)$. In terms of the whole vector $v(x) = (v^{(k)}(x), k = 1, ..., K)$, we have $v_{t+1}(x) = M(\delta(x))v_t(x)$, where $M(\delta(x)) = \text{diag}(M^{(k)}(\delta(x)), k = 1, ..., K)$ is the matrix representation of the local displacement $\delta(x)$.

### 3.5 DISENTANGLED ROTATIONS

The linear transformations of the sub-vectors $v^{(k)}(x)$ can be considered as rotations. Here we use the word "rotation" in the loose sense without strictly enforcing $M^{(k)}(\delta)$ to be orthogonal. $v(x)$ is like a multi-arm clock, with each arm $v^{(k)}(x)$ rotated by $M^{(k)}(\delta(x))$. The rotations of $v^{(k)}(x)$ for different $k$ and $x$ are disentangled. Here disentanglement means that the rotation of a sub-vector does not depend on other sub-vectors.

The disentanglement between different positions $x$ is the key feature of our model. Recall the change of image $\mathbf{I}$ is caused by the displacement of pixels, yet the rotations of sub-vectors $v^{(k)}(x)$ at different pixels $x$ are disentangled. This enables the agent to sense the displacement of a pixel only by sensing the rotations of the sub-vectors at this pixel without having to establish the correspondences between the pixels of consecutive frames.

### 3.6 PARAMETRIZATION

We can discretize the displacement $\delta(x)$ into a finite set of possible values $\{\delta\}$, and we learn a separate $M^{(k)}(\delta)$ for each $\delta$. We can also learn a parametric version of $M^{(k)}(\delta)$ as the second order Taylor expansion of a matrix-valued function of $\delta = (\delta_1, \delta_2)$, $M^{(k)}(\delta) = I + B_1^{(k)}\delta_1 + B_2^{(k)}\delta_2 + B_{11}^{(k)}\delta_1^2 + B_{22}^{(k)}\delta_2^2 + B_{12}^{(k)}\delta_1\delta_2$, where $I$ is the identity matrix, and $B^{(k)} = (B_1^{(k)}, B_2^{(k)}, B_{11}^{(k)}, B_{22}^{(k)}, B_{12}^{(k)})$ are matrices of coefficients of the same dimensionality as $M^{(k)}(\delta)$.

### 3.7 LOCAL MIXING

If $\delta(x)$ is large, $v_{t+1}^{(k)}(x)$ may contain information from adjacent image patches of $\mathbf{I}_t$ in addition to $\mathbf{I}_t[x]$. We can generalize the motion model in Equation (4) to allow local mixing of encoded vectors. Let $\mathcal{S}$ be a local support centered at 0. We assume that

$$v_{t+1}^{(k)}(x) = \sum_{dx \in \mathcal{S}} M^{(k)}(\delta(x), dx)v_t^{(k)}(x + dx) \tag{5}$$

In the learning algorithm, we discretize $dx$ and learn a separate $M^{(k)}(\delta, dx)$ for each $dx$.

## 4 LEARNING AND INFERENCE

The input data consist of the triplets $(\mathbf{I}_t, (\delta(x), x \in D_-), \mathbf{I}_{t+1})$, where $(\delta(x))$ is the given displacement field. The learned model consists of matrices $(W^{(k)}, M^{(k)}(\delta), k = 1, ..., K, \delta \in \Delta)$, where $\Delta$ is the range of $\delta$. In the case of parametric $M^{(k)}$, we learn the $B$ matrices in the second order Taylor expansion in subsection 3.6.

### 4.1 LOSS FUNCTIONS FOR LEARNING

We use the following loss functions:

(1) Rotation loss

$$L_{1,x,k} = \left\| W^{(k)} \mathbf{I}_{t+1}[x] - M^{(k)}(\delta(x)) W^{(k)} \mathbf{I}_t[x] \right\|^2. \tag{6}$$

For local mixing generalization, $L_{1,x,k} = \left\| W^{(k)} \mathbf{I}_{t+1}[x] - \sum_{\mathrm{d}x \in \mathcal{S}} M^{(k)}(\delta(x), \mathrm{d}x) W^{(k)} \mathbf{I}_t(x + \mathrm{d}x) \right\|^2$.

(2) Reconstruction loss

$$L_2 = \left\| \mathbf{I}_t - \sum_{x \in D_-} W^\top W \mathbf{I}_t[x] \right\|^2 + \left\| \mathbf{I}_{t+1} - \sum_{x \in D_-} W^\top W \mathbf{I}_{t+1}[x] \right\|^2. \tag{7}$$

In the learning algorithm, we learn the model by a weighted sum of the expectations of $\sum_{k=1}^{K} \sum_{x \in D_-} L_{1,x,k}$ and $L_2$, where the expectations are taken over the training pairs of images and the corresponding displacement fields.

### 4.2 INFERENCE OF MOTION

After learning $(W^{(k)}, M^{(k)}(\delta), \forall k, \forall \delta)$, for a testing pair $(\mathbf{I}_t, \mathbf{I}_{t+1})$, we can infer the pixel displacement field $(\delta(x), x \in D_-)$ by minimizing the rotation loss: $\delta(x) = \arg\max_{\delta \in \Delta} L_{1,x}(\delta)$, where

$$L_{1,x}(\delta) = \sum_{k=1}^{K} \left\| W^{(k)} \mathbf{I}_{t+1}[x] - M^{(k)}(\delta) W^{(k)} \mathbf{I}_t[x] \right\|^2 = \| W \mathbf{I}_{t+1}[x] - M(\delta) W \mathbf{I}_t[x] \|^2. \tag{8}$$

This algorithm is efficient because it can be parallelized for all $x \in D_-$ and for all $\delta \in \Delta$.

If we learn a parametric model for $M^{(k)}(\delta)$, we can infer the displacement field $(\delta(x), \forall x)$ by minimizing $\sum_x L_{1,x}(\delta(x))$ using gradient descent with an initialization of $(\delta(x))$ from random small values. To encourage the smoothness of the displacement field, we can add the penalty term $\| \nabla \delta(x) \|^2$.

### 4.3 BIOLOGICAL INTERPRETATIONS OF CELLS AND SYNAPTIC CONNECTIONS

The learned $(W^{(k)}, M^{(k)}(\delta)), \forall k, \delta$ can be interpreted as synaptic connections. For each $k$, $W^{(k)}$ corresponds to one set of connection weights. Suppose $\delta \in \Delta$ is discretized, then for each $\delta$, $M^{(k)}(\delta)$ corresponds to one set of connection weights, and $(M^{(k)}(\delta), \delta \in \Delta)$ corresponds to multiple sets of connection weights. After computing $v_{t,x}^{(k)} = W^{(k)} \mathbf{I}_t[x]$, $M^{(k)}(\delta) v_{t,x}^{(k)}$ is computed simultaneously for every $\delta \in \Delta$. Then $\delta(x)$ is inferred by max pooling according to Equation (8).

$v_{t,x}^{(k)}$ can be interpreted as activities of simple cells, and $\| v_{t,x}^{(k)} \|^2$ can be interpreted as activity of a complex cell. If we enforce norm stability so that $\| v_{t,x}^{(k)} \| \approx \| v_{t+1,x}^{(k)} \|$, then the complex cell response is invariant to the local motion and is related to the slowness property (Hyvärinen et al., 2003; Wiskott & Sejnowski, 2002), which is a by-product of our model if $M^{(k)}(\delta)$ is a rotation matrix, which is covariant with the local motion.

## 4.4 SPATIOTEMPORAL FILTERS AND RECURRENT IMPLEMENTATION

If we enforce norm stability or the orthogonality of $M^{(k)}(\delta)$, then minimizing $\|v_{t+1,x} - M(\delta)v_{t,x}\|^2$ over $\delta \in \Delta$ is equivalent to maximizing $\langle v_{t+1,x}, M(\delta)v_{t,x} \rangle$, which in turn is equivalent to maximizing $\|v_{t+1,x} + M(\delta)v_{t,x}\|^2$ so that $v_{t+1,x}$ and $M(\delta)v_{t,x}$ are aligned. This alignment criterion can be conveniently generalized to multiple consecutive frames, so that we can estimate the velocity at $x$ by maximizing the $m$-step alignment score $\|u\|^2$, where

$$u = \sum_{i=0}^{m} M(\delta)^{m-i} v_{t+i,x} = \sum_{i=0}^{m} M(\delta)^{m-i} W \mathbf{I}_{t+i}[x] \tag{9}$$

consists of responses of spatiotemporal filters, and $\|u\|^2$ corresponds to the energy of motion $\delta$ in the motion energy model (Adelson & Bergen, 1985) for direction selective cells. Thus our model is connected with the motion energy model. Moreover, our model enables a recurrent network for computing $u$ by $u_i = v_{t+i,x} + M(\delta)u_{i-1}$ for $i = 0, ..., m$, with $u_{-1} = 0$, and $u = u_m$. This recurrent implementation is much more efficient and biologically plausible than the plain implementation of spatiotemporal filtering which requires memorizing all the $\mathbf{I}_{t+i}$ for $i = 0, ..., m$. See (Pachitariu & Sahani, 2017) for a discussion of biological plausibility of recurrent implementation of spatiotemporal filtering in general.

## 5 EXPERIMENTS

First, in section 5.1 we introduce the datasets used to learn the models. Then in section 5.2 we show the learned Gabor-like units, and make connection with the spatial profile of simple cells in cat and Macaque monkey in terms of neuroscience metrics, indicating the biological plausibility of the learned units. Then in sections 5.3 and 5.4 we show the learned representations can be applied to infer displacement field reasonably well, and the representations can be trained either in a supervised or unsupervised manner. Please refer to appendix A for the implementation details. In appendices E and F we illustrate that the learned representations are capable of two extra tasks, frame animation and interpolation.

## 5.1 SYNTHETIC AND PUBLIC DATASETS

**V1Deform.** Usually it is difficult to get ground truth motions from natural video frames. Thus we consider learning from image pairs with synthetic motions. First we consider random smooth deformations for natural images. Specifically, We can obtain the training data by collecting static images for $(\mathbf{I}_t)$ and simulate the displacement field $(\delta(x))$. The simulated displacement field is then used to transform $\mathbf{I}_t$ to obtain $\mathbf{I}_{t+1}$. We retrieve natural images as $\mathbf{I}_t$ from MIT places365 dataset (Zhou et al., 2016). The images are scaled to $128 \times 128$. We sub-sample the pixels of images into a $m \times m$ grid ($m = 4$ in the experiments), and randomly generate displacements on the grid points, which serve as the control points for deformation. Then $\delta(x)$ for $x \in D$ can be obtained by spline interpolation of the displacements on the control points. We get $\mathbf{I}_{t+1}$ by warping $\mathbf{I}_t$ using $\delta(x)$ (Jaderberg et al., 2015). When generating a displacement $\delta = (\delta_1, \delta_2)$, both $\delta_1$ and $\delta_2$ are randomly sampled from a range of $[-6, +6]$. We generate $20,000$ pairs for training and $3,000$ pairs for testing. We name this dataset V1Deform.

**V1FlyingObjects.** Next we consider separating the displacement field into motions of the background and foreground, to jointly simulate the self-motion of the agent and the motion of the objects in the natural 3D scenes. To this end, we create a synthetic dataset, by applying affine transformations to background images collected from MIT places365 (Zhou et al., 2016) and foreground objects from a public 2D object dataset COIL-100 (Nene et al., 1996). The background images are scaled to $128 \times 128$, and the foreground images are randomly rescaled. To generate motion, we randomly sample affine parameters of translation, rotation, and scaling for both the foreground and background images. The motion of the foreground objects are relative to the background images, which can be explained as the relative motion between the moving object and agent. We tune the distribution of the affine parameters to keep the range of the displacement fields within $[-6, +6]$, which is consistent with the V1Deform dataset. Together with the mask of the foreground object and the sampled transformation parameters, we render the image pair $(\mathbf{I}_t, \mathbf{I}_{t+1})$ and its displacement field $(\delta(x))$ for each pair of background image and foreground image.

Specifically, we obtain the estimated masks from (tev, 2006) for the 2D foreground objects and remove some textureless objects, resulting in 96 objects with 72 views per object available. We generate $14,411$ synthetic image pairs with their corresponding displacement fields and further split $12,411$ pairs for training and $2,000$ pairs for testing. We name this dataset V1FlyingObjects. Compared with previous optical flow dataset like Flying Chairs (Dosovitskiy et al., 2015) and scene flow dataset like FlyingThings3D (Mayer et al., 2016), the proposed V1FlyingObjects dataset has various foreground objects with more realistic texture and smoother displacement fields, which simulates more realistic environments. We shall release this dataset.

**MPI-Sintel.** MPI-Sintel (Butler et al., 2012; Wulff et al., 2012) is a public dataset designed for the evaluation of optical flow derived from rendered aritificial scenes, with special attention to realistic image properties. Since MPI-Sintel is relatively small which contains around a thousand image pairs, we use it only for testing the learned models in the inference of displacement field and frame animation, as described in details in sections 5.3 and Appendix E. We use the final version of MPI-Sintel and resize each frame into size $128 \times 128$. We select frame pairs whose motions are within the range of $[-6, +6]$, resulting in 384 frame pairs in total.

**MUG Facial Expression.** MUG Facial Expression dataset (Aifanti et al., 2010) records natural facial expression videos of 86 subjects sitting in front of one camera. This dataset has no ground truth of displacement field, which we use for unsupervised learning as stated in details in section 5.4. 200 videos with 30 frames are randomly selected for training, and anther 100 videos are sampled for testing. The frame images are resized to $64 \times 64$.

## 5.2 LEARNED GABOR-LIKE UNITS WITH QUADRATURE PHASE RELATIONSHIP

In this section we show and analyze the learned units. Figure 8(a) displays the learned units, i.e., rows of $W^{(k)}$, on V1Deform. The units are learned with non-parametric $M(\delta)$, i.e., we learn a separate $M(\delta)$ for each displacement. $\delta(x)$ is discretized with an interval of $0.5$. Similar patterns can be obtained by using parametric version of $M(\delta)$. Please refer to the supplementary E and D for more results, including animation of filters, filters learned with local mixing motion model (eqn. (5)), with different block sizes, and learned on V1FlyingObjects. V1-like patterns emerge from the learned units. Moreover, within each sub-vector, the orientations and frequencies of learned units are similar, while the phases are different.

To further analyze the spatial profile of the learned units, we fit every unit by a two dimensional Gabor function (Jones & Palmer, 1987): $h(x', y') = A \exp(-(x'/\sqrt{2}\sigma_{x'})^2 - (y'/\sqrt{2}\sigma_{y'})) \cos(2\pi f x' + \phi)$, where $(x', y')$ is obtained by translating and rotating the original coordinate system $(x_0, y_0)$: $x' = (x - x_0)\cos\theta + (y - y_0)\sin\theta, y' = -(x - x_0)\sin\theta + (y - y_0)\cos\theta$. The fitted Gabor patterns are shown in figure 8(b), with the average fitting $r^2$ equal to $0.96$ (std $= 0.04$). The average spatial-frequency bandwidth is 1.13 octaves, with range of $0.12$ to $4.67$. Figure(c) shows the distribution of the spatial-frequency bandwidth, where the majority falls within range of $0.5$ to $2.5$. The characteristics are reasonably similar to those of simple-cell receptive fields in the cat (Issa et al., 2000) (weighted mean 1.32 octaves, range of $0.5$ to $2.5$) and the macaque monkey (Foster et al., 1985) (median 1.4 octaves, range of $0.4$ to $2.6$). To analyze the distribution of the spatial phase $\phi$, we follow the method in (Ringach, 2002) to transform the parameter $\phi$ into an effective range of $0$ to $\pi/2$, and plot the histogram of the transformed $\phi$ in figure 8(c). The strong bimodal with phases clustering near $0$ and $\pi/2$ is consistent with those of the macaque monkey (Ringach, 2002).

In the above experiment, we fix the size of the convolutional filters ($16 \times 16$ pixels). A more reasonable model is to have different sizes of convolutional filters, with small size filters capturing high frequency content and big size filters capturing low frequency content. For fixed size filters, they should only account for the image content within a frequency band. To this end, we smooth every image by two Gaussian smoothing kenels (kernel size 8, $\sigma = 1, 4$), and take the difference between the two smoothed images as the input image of the model. The effect of the two smoothing kernels is similar to a bandpass filter, so that the input images are constrained within a certain range of frequencies. The learned filters on V1Deform are shown in 3(a). Again for every unit, we fit it by a two dimensional Gabor function, resulting in an average fitting $r^2 = 0.83$ (std $= 0.12$). Following the analysis of (Ringach, 2002; Rehn & Sommer, 2007), a scatter plot of $n_x = \sigma_x f$ versus $n_y = \sigma_y f$ is constructed in Figure 3(b) based on the fitted parameters, where $n_x$ and $n_y$ represent the width and length of the Gabor envelopes measured in periods of the cosine waves. Compared to Sparsenet

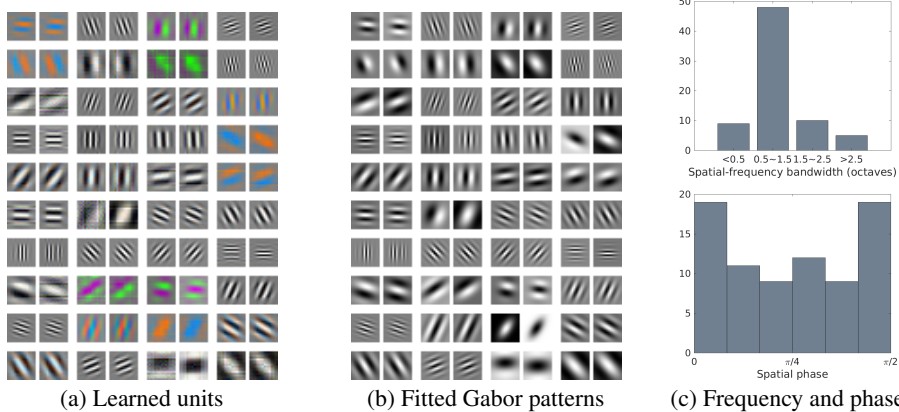

(a) Learned units      (b) Fitted Gabor patterns      (c) Frequency and phase

Figure 2: Learned results on V1Deform. (a) Learned units. Each block shows two learned units within the same sub-vector. (b) Fitted Gabor patterns. (c) Distributions of spatial-frequency bandwidth (in octaves) and spatial phase $\phi$.

(Olshausen & Field, 1996; 1997), the learned units by our model have more similar structure to the receptive fields of macaque monkey.

We also show profile of the learned units within each sub-vector in Figure 3(c). Within each sub-vector, the frequency $f$ and orientation $\theta$ of the paired units tends to be the same. More importantly, most of the paired units differ in phase $\phi$ by approximately $\pi/2$, consistent with the quadratic phase relationship between adjacent simple cells (Pollen & Ronner, 1981; Emerson & Huang, 1997).

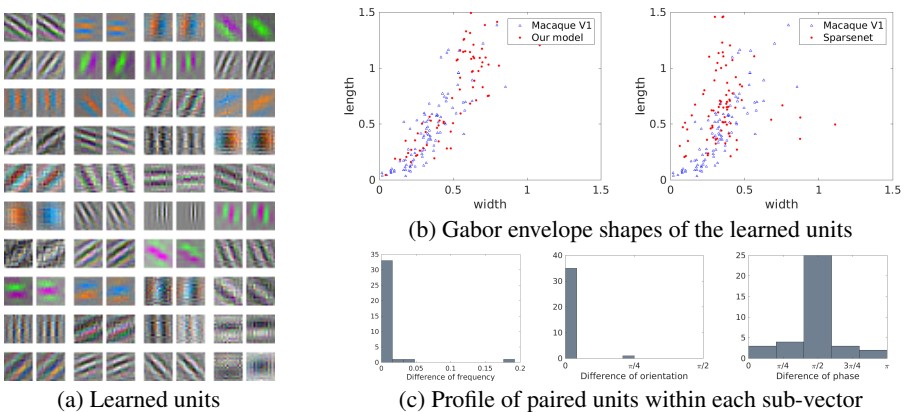

(a) Learned units      (c) Profile of paired units within each sub-vector

Figure 3: Learned results on band-pass image pairs from V1Deform. (a) Learned units. Each block shows two learned units within the same sub-vector. (b) Distribution of the Gabor envelope shapes in the width and length plane. (c) Difference of frequency $f$, orientation $\theta$ and phase $\phi$ of paired units within each sub-vector.

## 5.3 INFERENCE OF DISPLACEMENT FIELD

We then test the learned representations in terms of inferring the displacement field $(\delta(x))$ between pairs of frames $(\mathbf{I}_t, \mathbf{I}_{t+1})$. To get valid image patches for the inference, we leave out those displacements at image border (8 pixels at each side).

We infer the displacement field $(\delta(x))$ using the learned vector and matrix representation. On top of that, we also train a CNN model with ResNet blocks (He et al., 2016) to refine the inferred displacement field. In training this CNN, the input is the inferred displacement field, and the output is the ground truth displacement field, with least squares regression loss. The detailed model structure is in appendix I. For V1Deform, we train the representational model without refinement and test on the testing set of V1Deform. For V1FlyingObjects, we train both the representational model and the

refinement CNN on the training set, and test on the testing set of V1FlyingObjects and MPI-Sintel datasets. The refinement CNN is to approximate the processing in visual areas V2-V6 that integrates and refines the motion perception in V1 (Gazzaniga et al., 2002; Lyon & Kaas, 2002; Moran & Desimone, 1985; Born & Bradley, 2005; Allman & Kass, 1975).

Figure 4 displays several examples of the inferred displacement field, learned with non-parametric $M(\delta)$, using the local mixing motion model (eqn. (5)), where the local support $\mathcal{S}$ is in a range of $[-4, +4]$, and $\mathrm{d}x$ is taken with a sub-sampling rate of 2. We also show the inferred results from pre-trained FlowNet 2.0 (Ilg et al., 2017) model as a comparison. In Table 1, we report the average endpoint error (EPE) of the inferred results. We compare with some baseline methods, such as the FlowNet and its variants (Dosovitskiy et al., 2015; Ilg et al., 2017), by obtaining the pre-trained models and testing on the corresponding datasets. Note that those methods train deep and complicated neural networks with large scale datasets to predict optical flows in supervised manners, while our model can be treated as a simple one-layer auto-encoder network, accompanied by weight matrices representing motions. We achieve competitive results to these methods.

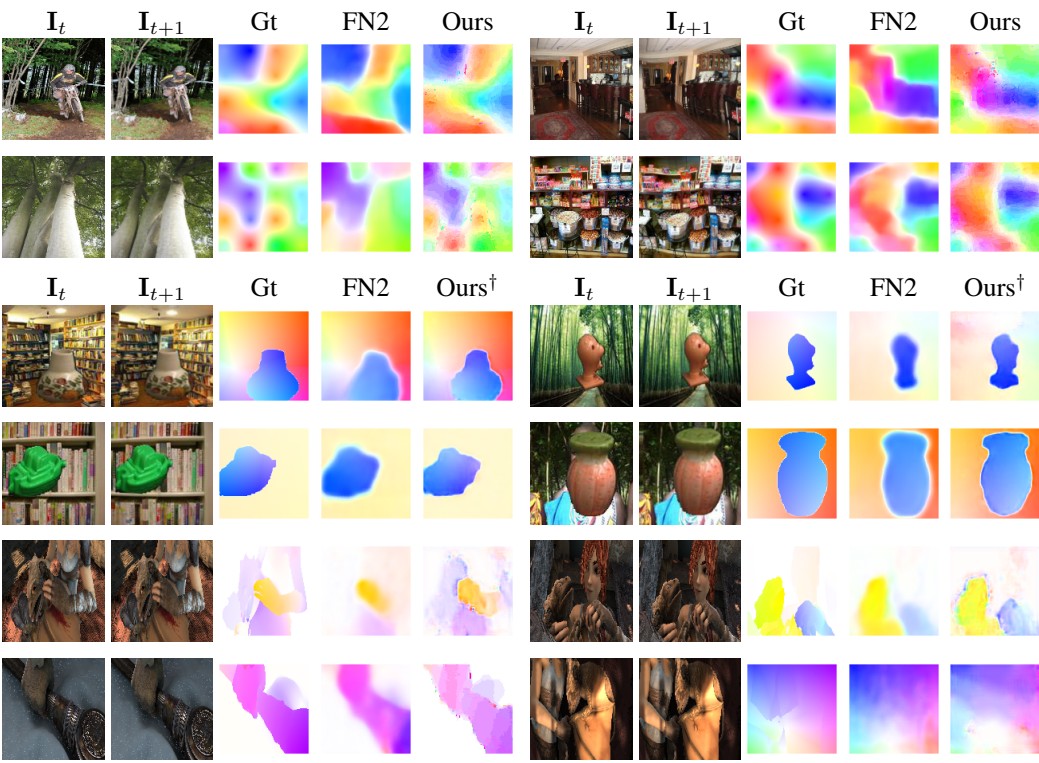

Figure 4: Examples of inference of displacement field on V1Deform, V1FlyingObjects and MPI-Sintel. For each block, from left to right are $\mathbf{I}_t$, $\mathbf{I}_{t+1}$, ground truth displacement field and inferred displacement field by pre-trained FlowNet 2.0 model and our learned model respectively. [†] indicates that the results are refined by the refinement CNN. The displacement fields are color coded. See supplementary for the color code (Liu et al., 2010).

Table 1: Average endpoint error of the inferred displacement. (FN stands for FlowNet)

|  | FN-C | FN-S | FN-CS | FN-CSS | FN2 | Ours | Ours + Refine |
|---|---|---|---|---|---|---|---|
| V1Deform | 1.324 | 1.316 | 0.713 | 0.629 | 0.686 | **0.444** | - |
| V1FlyingObjects | 0.852 | 0.865 | 0.362 | 0.299 | 0.285 | 0.442 | **0.202** |
| MPI-Sintel | 0.363 | 0.410 | 0.266 | 0.234 | **0.146** | 0.337 | 0.212 |

## 5.4 UNSUPERVISED LEARNING

Assume there is a dataset of frame sequences, where the ground truth displacement fields are unknown. We can learn the model by the following steps: (1) first we take the frames as static images, deform the images like what we did for V1Deform to get image pairs, and learn the model as an initialization; (2) then we infer the displacement fields between adjacent pair of frames using the initialized model; (3) using adjacent pair of frames as training data, we alternatively update the model parameters and re-infer displacement fields. In this task, we use the parametric $M$ and infer the displacement field by gradient descent on a weighted sum of $\sum_x L_{1,x}(\delta(x))$ and $\|\nabla\delta(x)\|^2$. At each iteration, we start the inference from the inferred displacement field from the last iteration.

We test the unsupervised learning on MUG Facial Expression dataset (Aifanti et al., 2010). Figure 5 shows some examples of inferred displacement fields by the unsupervised learning. The inference results are reasonable, which capture the motions around eyes, eyebrows, chin or mouth. See supplementary D and H for the learned filters and more inferred examples.

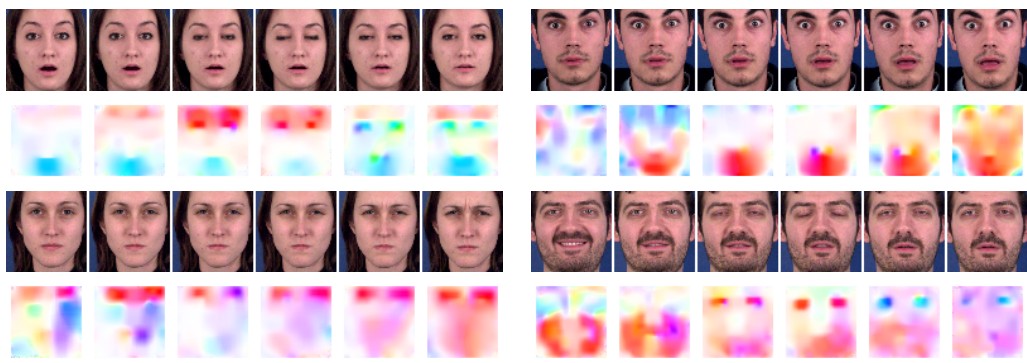

Figure 5: Examples of inferred displacement fields by unsupervised learning. The top row shows the observed image sequences, while the bottom row shows the inferred color coded displacement field (Liu et al., 2010).

We perform ablation studies to analyze the effect of two components of the proposed model: (1) dimensionality of sub-vectors; (2) sub-sampling rate. Please refer to supplementary G for the details.

## 6 CONCLUSION

This paper proposes a simple representational model that couples vector representations of local image contents and matrix representations of local motions. Unlike existing models for V1 that focus on statistical properties of natural images or videos, our model serves a direct purpose of perception of local motions caused by the relative motions between the agent and the 3D environment. Our model learns Gabor-like units with quadrature phases. We also give biological interpretations of the learned model and connect it to the spatiotemporal energy model. Our model is novel, and it is our hope that it adds to our understanding of motion perception in V1 in terms of modeling and inference.

In our future work, we shall study the inference of ego-motion, object motions and 3D depth information based on local pixel displacements by expanding our model. We shall also extend our model to stereo in binocular vision by allowing separate encoding matrices for the pair of input images to the two eyes related by pixel displacements caused by depths.

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

## A    IMPLEMENTATION DETAILS

We learn our model $(W^{(k)}, M^{(k)}(\delta), k = 1, ..., K)$ from image pairs $(\mathbf{I}_t, (\delta(x)), \mathbf{I}_{t+1})$. The number of sub-vectors $K = 40$, and the number of units in each sub-vector $v^{(k)}(x)$ is 2. We also try other dimensionalities of sub-vector, e.g., 4 and 6. See supplementary materials. Each row of the encoding matrix $W^{(k)}$ is a filter. The size of the filter is $16 \times 16$, with a sub-sampling rate of 8 pixels in order to get $D_-$. We learn the model using stochastic gradient descent implemented by Adam (Kingma & Ba, 2014), with learning rate $0.0008$.

For unsupervised learning in section 5.4, since the image size reduces to 64, we use kernel size 8 with a sub-sampling rate of 4 pixels. In stage (1) for model initialization, we set the range of displacement to $[-3, +3]$. Displacements at image border are left out.

## B    COLOR CODE OF DISPLACEMENT FIELD

Figure 6 shows the color map for the color coded displacement fields used in this paper (Liu et al., 2010).

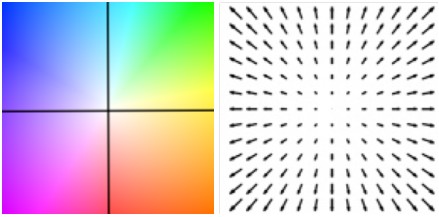

Figure 6: Color map for the color coded displacement fields. The displacement of every pixel in this map is the vector from the center of the square to this pixel. The center pixel does not move. The range of color is taken according to the maximum length of flows in each displacement field.

## C    ANIMATION OF LEARNED UNITS: MOVING V1-LIKE UNITS

We have $M^{(k)}(\delta)v_t^{(k)}(x) = M^{(k)}(\delta)W^{(k)}\mathbf{I}[x]$, where each row of the encoding matrix $W^{(k)}$ serves as a filter. Let $W^{(k)}(\delta) = M^{(k)}(\delta)W^{(k)}$. By changing values of $\delta$, we can animate $W^{(k)}$ to make it move. Figure 7 shows several examples of the animation. Each block shows a certain $W^{(k)}$ animated by a fixed $\delta$. Each column shows the units in the same $W^{(k)}(\delta)$. As $\delta$ changes, the orientations of learned units remain the same, while the phases change, and the units belonging to the same sub-vector tend to have similar movements.

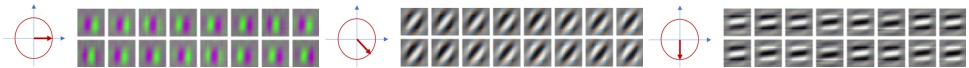

Figure 7: Animation of the learned filters

## D  LEARNED FILTERS

Figure 8 shows the learned filters under different settings, including learned on V1FlyingObjects, learned with parametric $M$, learned with local mixing model (eqn 5) and learned unsupervisedly.

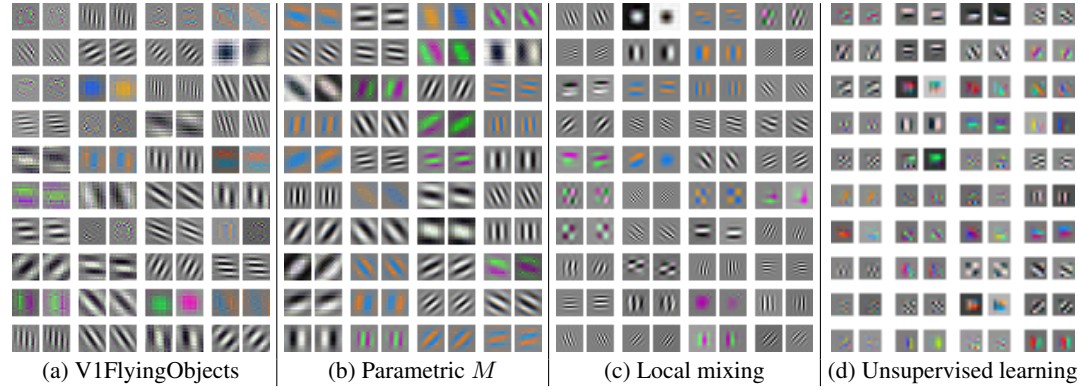

(a) V1FlyingObjects    (b) Parametric $M$    (c) Local mixing    (d) Unsupervised learning

Figure 8: Filters learned from under different settings: (a) filters learned on V1FlyingObjects with non-parametric $M$; (b) filters learned with parametric $M$; (c) filters learned with non-parametric $M$ and local mixing motion model (model used in section 5.3); (d) filters learned on MUG Facial expression dataset with unsupervised learning (model used in section 5.4).

## E  MULTI-STEP FRAME ANIMATION

Given the starting frame $\mathbf{I}_0(x)$ and a sequence of displacement fields $\{\delta_1(x), ..., \delta_T(x), \forall x\}$, we can animate the subsequent multiple frames $\{\mathbf{I}_1(x), ..., \mathbf{I}_T(x)\}$ using the learned model. We use the model with local mixing with the same setting as in section 5.3. We introduce a re-encoding process when performing multi-step animation. At time $t$, after we get the next animated frame $\mathbf{I}_{t+1}$, we take it as the observed frame at time $t + 1$, and re-encode it to obtain the latent vector $v_{t+1}$ at time $t + 1$.

Figure 9 displays several examples of a 6-step animation, learned with non-parametric version of $M$ on V1Deform and V1FlyingObjects. The animated frames match the ground truth frames well. As a quantitative evaluation, we compute the per pixel distance between the predicted frames and observed frames, which is $9.032$ in the testing dataset for V1Deform and $12.076$ for V1FlyingObjects.

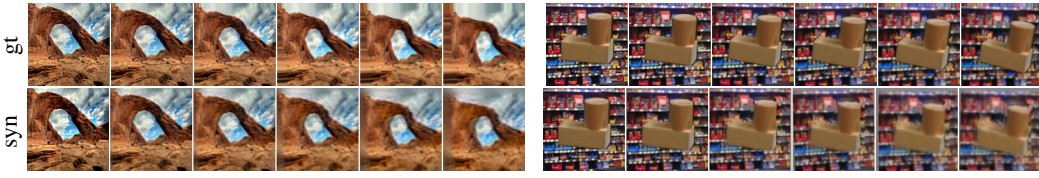

Figure 9: Examples of multi-step animation. For each block, the first row shows the ground truth frame sequences, while the second row shows the animated frame sequences.

## F  FRAME INTERPOLATION

Inspired by the animation and inference results, we show that our model can also perform frame interpolation, by combining the animation and inference together. Specifically, given a pair of starting

frame $\mathbf{I}_0$ and end frame $\mathbf{I}_T$, we want to derive a sequence of frames $(\mathbf{I}_0, \mathbf{I}_1, ..., \mathbf{I}_{T-1}, \mathbf{I}_T)$ that changes smoothly. Let $v_0(x) = W\mathbf{I}_0[x]$ and $v_T(x) = W\mathbf{I}_T[x]$ for each $x \in D$. At time step $t+1$, like the inference, we can infer displacement field $\delta_{t+1}(x)$ by

$$\hat{v}_{t+1}^{(k)}(x, \delta) = \sum_{\mathrm{d}x \in \mathcal{S}} M^{(k)}(\delta, \mathrm{d}x) v_t^{(k)}(x + \mathrm{d}x), \forall x \in D, \forall \delta \in \Delta, \forall k \tag{10}$$

$$\delta_{t+1}(x) = \arg\min_{\delta \in \Delta} \sum_{k=1}^{K} \left\| v_T^{(k)} - \hat{v}_{t+1}^{(k)}(x, \delta) \right\|^2, \forall x \in D \tag{11}$$

Like the animation, we get the animated frame $\mathbf{I}_{t+1}$ by decoding $\hat{v}_{t+1}(x, \delta_{t+1}(x))$, and then re-encode it to obtain the latent vector $v_{t+1}(x)$.

The algorithm stops when $\mathbf{I}_t$ is close enough to $\mathbf{I}_T$ (mean pixel error $< 10$). Figure 10 shows several examples, learned with non-parametric $M$ on V1Deform and V1FlyingObjects. For 96.0% of the testing pairs, the algorithm can accomplish the frame interpolation within 10 steps. With this algorithm, we are also able to infer displacements larger than the acceptable range of $\delta$.

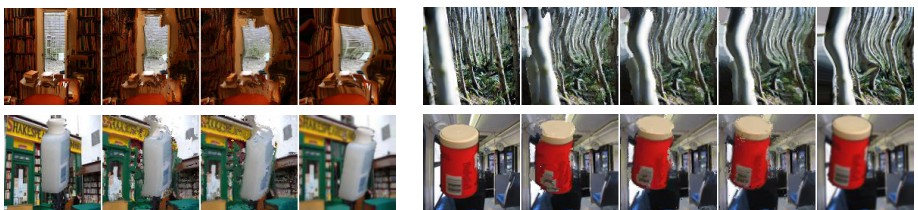

Figure 10: Examples of frame interpolation, learned with non-parametric $M$. For each block, the first frame and last frame are given, while the frames between them are interpolated frames.

## G  ABLATION STUDY

We perform an ablation study to analyze the effect of several components of the proposed model. All the models in the ablation study are trained with non-parametric $M(\delta)$ on V1Deform.

**Dimensionality of sub-vectors.** In the experiments, we assume that the number of units in each sub-vector $v^{(k)}(x)$ is 2, so that within each sub-vector, a pair of V1-like patterns are learned. However, we show that the dimensionality of sub-vectors does not have to be 2. In figure 11(a) we show the learned filters with dimension of sub-vectors equal to 4 and 6. For fair comparison, we fix the total number of units in the whole vector to 96, and change the number of units in each sub-vector. Table 2 summarizes the quantitative analysis of the models learned with different dimensionalities of sub-vectors, in terms of the performances of multi-step animation and inference of displacement field. As the dimensionality of sub-vectors increases, the error rates of the two tasks decrease first and then increase. Besides, in figure 11(b) we show the learned filters without the assumption of sub-vectors.

Table 2: Quantitative analysis of the models learned with different dimensionalities of sub-vectors.

| Sub-vector dim | 2 | 4 | 6 | 8 | 12 |
|---|---|---|---|---|---|
| animation MSE | 8.684 | 8.387 | 7.486 | 7.926 | 8.412 |
| inference EPE | 0.554 | 0.520 | 0.496 | 0.500 | 0.528 |

**Sub-sampling rate.** Another factor that may affect the learned model is the sub-sampling rate in order to get $D_-$. In the experiments, we use sub-sampling rate 8, which is half of the filter size. We can also increase or decrease the sub-sampling rate to make the adjacent image patches connected with each other more loosely or tightly. Table 3 summarizes the performance of learned models with different sub-sampling rates, in terms of multi-step animation and inference of displacement field.

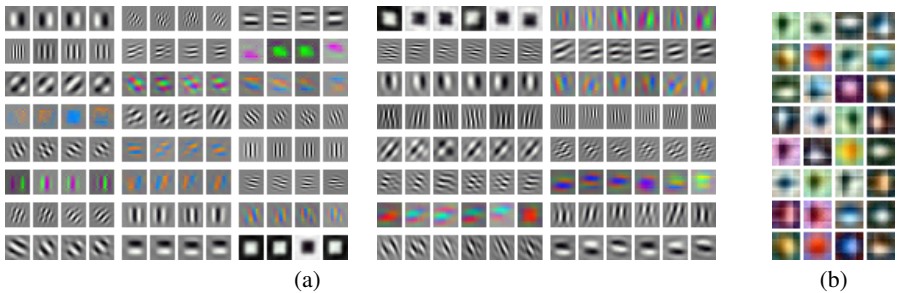

(a)                     (b)

Figure 11: (a) Filters learned with higher dimension of sub-vectors. The total number of units in the whole vector is fixed to 96. Each block shows the learned units within the same sub-vectors. (b) Filters learned without sub-vector assumption.

Table 3: Quantitative analysis of the models learned with different sub-sampling rates.

| Sub-sampling rate | 4 | 8 | 16 |
|---|---|---|---|
| animation MSE | 7.492 | 8.094 | 10.808 |
| inference EPE | 0.658 | 0.505 | 0.565 |

## H    UNSUPERVISED LEARNING: MORE RESULTS

See figure 12 for more inference results by unsupervised learning.

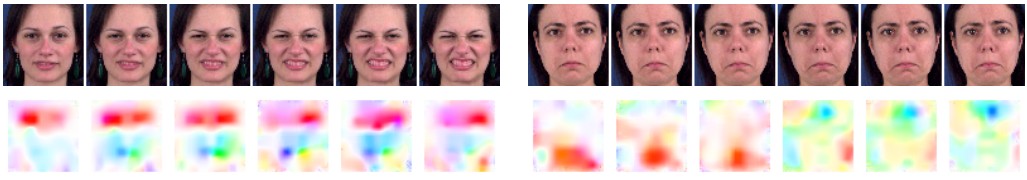

Figure 12: More examples of inferred displacement fields by unsupervised learning. The top row shows the observed image sequences, while the bottom row shows the inferred color coded displacement field.

## I    REFINEMENT CNN MODEL STRUCTURE

Table 4: Refinement CNN model structure

| |
|---|
| $3 \times 3$ conv. 8 ReLU, stride 1 |
| $3 \times 3$ conv. 16 ReLU, stride 1 |

| |
|---|
| Residual blocks |
| $\left\{ \begin{array}{l} 3 \times 3 \text{ conv. 16 \ BN \ ReLU, \ stride 1} \\ 3 \times 3 \text{ conv. 16 \ BN, \ stride 1} \end{array} \right\} \times 4$ |

| |
|---|
| $3 \times 3$ conv. 8 ReLU, stride 1 |
| $3 \times 3$ conv. 2 ReLU, stride 1 |

