# OpenReview forum: "Learning vector representation of local content and matrix representation of local motion, with implications for V1"
_ICLR.cc/2020/Conference — Reject_

### Official Review · AnonReviewer1 · 2019-10-21
**Official Blind Review #1**

**Rating:** 3

**Review:**

Summary:
This paper proposes a representation model for describing local pixel displacement. The proposed model uses matrix multiplication for optical flow estimation, where an image is transformed into a vector and the local motion is modeled by a matrix.
The recommendation of this work is based on the following reasons. First, the motivation of the proposed method is not convincing. While the proposed ideas are interesting, it is not clear why this approach sheds light on our understanding of motion perception. Is there any psychological evidence to support the proposed model? Or the authors simply take some ideas form V1 model and add a module to “explain” motion?  Second, the experimental results are not sufficient to demonstrate the effectiveness of the proposed model.
Major issues:
First, while it is interesting to use matrix multiplication to model motion, it is not clear why the motion between patches I_t[x] and I_{t+1}[x] can be approximated with linear transformation (Section 3.4). Furthermore, it is not clear why the transformation M only depends on the displacement of the center pixel whereas different pixels in a patch I_t[x] could have different displacements.
Second, the proposed model for optical flow estimation is only evaluated on the proposed V1Deform dataset. If the authors position this paper “may shed light on our motion perception in primary visual cortex”, the authors certainly need to carry out sufficient psychophysical experiments.
Minor issues:
First, Eq. 2 does not seem correct to me. The left and right sides of Eq. 2 have different dimensions.
Second, the authors may consider using {} instead of () to define a set of pixels or vectors in Section 3.1.
Third, while the reconstruction loss (Eq. 7) is used in this paper, I wonder what the results would be like if the authors simply enforce W’W=I instead.


**Experience Assessment:**

I have published in this field for several years.

**Review Assessment: Checking Correctness Of Derivations And Theory:**

I assessed the sensibility of the derivations and theory.

**Review Assessment: Checking Correctness Of Experiments:**

I assessed the sensibility of the experiments.

**Review Assessment: Thoroughness In Paper Reading:**

I read the paper at least twice and used my best judgement in assessing the paper.

---

> ### Author Response · Authors · 2019-11-15
> **Response to Reviewer #1 (Part 1)**
>
> Thanks for your valuable comments and suggestions.
>
> Q1: “it is not clear why this approach sheds light on our understanding of motion perception. Is there any psychological evidence to support the proposed model?”
>
> A1: In this paper, we seek to explain two important features of the simple cells in V1. One is that they can be approximated by Gabor filters. The other is that adjacent cells have quadrature phase relation. Our motion model gives simple explanations to the above two features.
>
> About motion perception, we did consult experts on the neuroscience and psychophysics of motion perception in V1. Existing neuroscience models are usually based on the spatial-temporal filters, such as the motion energy model of [1]. In Subsection 4.4, we connect our work to this model to explain the emergence of spatial-temporal filters. Moreover, we present a recurrent implementation of the spatial-temporal filtering. This recurrent implementation is more efficient and more biologically plausible than plain implementation of spatial-temporal filters which requires memorizing the past frames.
>
> [1] Edward H Adelson and James R Bergen. Spatiotemporal energy models for the perception of motion. Josa a, 2(2):284–299, 1985.
>
> In our paper, we also follow the protocol of the neuroscience papers [2,3] to evaluate the learned filters.
>
> [2] Dario L Ringach. Spatial structure and symmetry of simple-cell receptive fields in macaque primary visual cortex. Journal of neurophysiology, 88(1):455–463, 2002.
>
> [3] Martin Rehn and Friedrich T Sommer. A network that uses few active neurons to code visual input predicts the diverse shapes of cortical receptive fields. Journal of computational neuroscience, 22(2):135–146, 2007.
>
> Q2: “why the motion between patches I_t[x] and I_{t+1}[x] can be approximated with linear transformation”
>
> A2: Thanks for the insightful question. One motivation of our model is based on Fourier analysis: An image patch I can be expressed as I(x) = sum_k c_k e^{i<\omega_k, x>} in a Fourier decomposition. If we shift it by dx, the shifted image patch J(x) = I(x - dx) = sum_k c_k e^{-i<\omega_k, dx>} e^{i<\omega_k, x>}. The change from the complex number c_k to c_k e^{-i<\omega_k, dx>} corresponds to rotating a 2D vector by a 2 x 2 matrix. This is a simple example that the shift can be represented by a linear transformation in the frequency domain, as a change in phase.
>
> We want to emphasize that our model does not assume Fourier basis or its localized version such as Gabor filters. Our model figures it out with generic vector and matrix representations.
>
> We have added the above motivation to the introduction.
>
> Q3: “why the transformation M only depends on the displacement of the center pixel whereas different pixels in a patch I_t[x] could have different displacements”
>
> A3: Thanks for the good question. We assume that the motion is smooth, so that within a relative small local patch, the motion is constant. Of course, the patch size or the filter size should be related to image resolution. For images with higher resolution, we may want to use smaller filter size to make this assumption hold. We have added a comment on this point in the introduction when discussing the Fourier analysis motivation.
>
> To be continued in the next message.

---

> > ### Author Response · Authors · 2019-11-15
> > **Response to Reviewer #1 (Part 2)**
> >
> >
> > Q4: “the motivation of the proposed method”, “Or the authors simply take some ideas form V1 model and add a module to “explain” motion? “
> >
> > A4: One motivation is based on Fourier analysis as mentioned above. Please see our answer to Q2. Another motivation is from previous papers that use matrices to represent camera motion or self-motion. Specifically, in [1], the authors study the change of images when the camera undergoes motion. Each image frame is represented by a vector. The camera motion is represented by a matrix. This idea was also alluded to in [2]. In [3], the authors study the grid cells as forming a high-dimensional vector representation of the 2D position of the agent. The self-motion of the agent is represented by a matrix.
> >
> > [1] Jayaraman, Dinesh, and Kristen Grauman. "Learning image representations tied to ego-motion." Proceedings of the IEEE International Conference on Computer Vision. 2015.
> >
> > [2] Paccanaro, Alberto, and Geoffrey E. Hinton. "Learning distributed representations of concepts using linear relational embedding." IEEE Transactions on Knowledge and Data Engineering 13.2 (2001): 232-244.
> >
> > [3] Gao, Ruiqi, et al. "Learning grid cells as vector representation of self-position coupled with matrix representation of self- motion." Seventh International Conference on Learning Representations (2019).
> >
> > Adding a motion module on top of existing V1 models was not a motivation of our work. But indeed our model relates the linear representations of consecutive image frames, so that our model complements existing models based on linear representations. Moreover, our motion model makes the concept of sub-vectors explicit, in the sense that the sub-vectors are what are rotated by the matrices.
> >
> > Q5: “the experimental results are not sufficient to demonstrate the effectiveness of the proposed model.”
> >
> > A5: To strengthen the experiments, we have added experiments on two more datasets. Please see Subsections 5.1, 5.3 and Appendices E, F for details. These experiments show that our method achieves competitive performances on optical flow estimation.
> >
> > Q6: About minor issues
> >
> > A6: Thanks for your careful reading.
> >
> > (1) Thanks for pointing out the issue with the notation of Eq. 2. We now use the more generic notation I = {\bf W}^T V in Eq. 2. Otherwise, we should have used I = \sum_x W^T(x) v(x), where each column of W^T(x) is of the same dimension as I, where we translate the filters W to pixel x and zero-padding the pixels outside the filters.
> >
> > (2) We have changed the notation in Subsection 3.1 following your suggestion.
> >
> > (3) We have tried to replace the reconstruction loss with |I – W’W|^2. However, the learned filters have no obvious pattern. W’W = I is a stricter constraint than the reconstruction loss, since it requires the reconstruction to hold for any I, whereas in the reconstruction loss, we only want the reconstruction to work for natural images, that is, the learned W captures statistical properties of natural images.

---

### Official Review · AnonReviewer5 · 2019-11-02
**Official Blind Review #5**

**Rating:** 1

**Review:**

The authors propose a model for learning local pixel motions between pairs of frames using local image representations and relative pixel displacements between agents and objects.  The model learned is compared to the ability of the primary visual cortex where adjacent simple cells share quadrature relationships and capture local motion.

"The representation theory underlies much of modern mathematics and holds the key to the quantum
theory (Zee, 2016)."
Can the relevance of this claim be elaborated on?

"Figure 1 illustrates the scheme of representation."
Please provide more detail here on what is happening in the figure.  The caption and reference here are not informative to what the figure is representing.

"We obtain the training data by collecting static images for (It) and simulate the
displacement field ...  We refer to this method as self-supervised learning"
This is not self-supervised learning.  In self-supervised learning the training label/signal is generated by the system.  In this case artificial data is being generated as the displacement between images is sampled.

Since the motion between images is artificially generated what guarantees are there that the model is learning to capture realistic motion behavior?  Why not use adjacent video frames?

"Note that those methods train deep and complicated neural networks with large scale datasets to
predict optical flows in supervised manners, while our model can be treated as a simple one-layer
network, accompanied by weight matrices representing motions."
Is there a comparison on execution times of the different approaches?

"by obtaining the pre-trained models and testing on V1Deform testing data"
Is this a fair comparison if the proposed approach was trained on V1Deform training data and the comparison methods were not.  A more appropriate comparison would be to apply all the methods to infer the displacement fields between video frames which is also a more natural application.  This can be controlled to contain small motions if needed.  Why nt use the MUG dataset here?

"Displacements at image border are leaved out" -> left out

Sections 5.4, 5.5 and 5.6 show only qualitative results with no comparison methods.  Can the authors provide reasons that other methods could not be used for evaluation?

I am not sure I understand the motivation for the approach.  Why do we need this over other methods that can better capture larger motions.  This needs to be more clear from the introduction.  Why do we care if the approach captures aspects of V1 for the tasks presented?

The work is sensible and the approach is clear but I found the evaluation and motivation lacking in key areas that I mention above.  The authors should revise and make it clear to the reader why we should care about this problem.  Aligning with V1 is interesting but it does not come into play in the applications of the approach or the analysis so I am not sure why I should care.  The evaluation also needs to be much more convincing before I could recommend acceptance.

**Experience Assessment:**

I have published one or two papers in this area.

**Review Assessment: Checking Correctness Of Derivations And Theory:**

I assessed the sensibility of the derivations and theory.

**Review Assessment: Checking Correctness Of Experiments:**

I assessed the sensibility of the experiments.

**Review Assessment: Thoroughness In Paper Reading:**

I read the paper thoroughly.

---

> ### Author Response · Authors · 2019-11-15
> **Response to Reviewer #5 (Part 1)**
>
> Thank you for your valuable comments and suggestions.
>
> Q1: “‘The representation theory underlies much of modern mathematics and holds the key to the quantum theory (Zee, 2016).’ Can the relevance of this claim be elaborated on?”
>
> A1: Yes. In representation theory in mathematics, for a group G, each element g is represented by a matrix M(g) acting on the vector v in a vector space. For two elements g1 and g2 in G, g1*g2 is represented by M(g1) M(g2). In our work, the displacements dx form a 2D Euclidean group. Each dx is represented by a matrix M(dx) acting on the vector v(x) that represents the local image content.
>
> In quantum physics, a particle at position x is represented by a vector v(x) in a Hilbert space. If the particle undergoes a displacement dx, the vector is transformed by a displacement matrix (or operator) M(dx), so that v(x+dx) = M(dx) v(x). In our work, v(x) represents the local image content, and M(dx) represents pixel displacement.
> More generally, a particle of a certain momentum with a certain spin (as well as other properties) is represented by a vector in a Hilbert space. When the particle undergoes a Lorentz transformation (a more general notion of displacement in space-time), the vector is multiplied by a matrix (or operator) representing the Lorentz transformation. Different types of particles correspond to different schemes of representing the Lorentz transformations.
> We adopt such mathematical language in our work. More generally, we may use vectors to represent various objects in the image, and use matrices to represent the motions of these objects.
> Such a mathematical language was adopted by earlier papers before.
> In [1], the authors study the change of images when the camera undergoes motion. Each image frame is represented by a vector. The camera motion is represented by a matrix. This idea was also alluded to in [2].
> In [3], the authors study the grid cells as forming a high-dimensional vector representation of the 2D position of the agent. The self-motion of the agent is represented by a matrix.
> Unlike vector representation that is common in deep learning models, the matrix representation is relatively rare. Our work is an example along this theme.
>
> [1] Jayaraman, Dinesh, and Kristen Grauman. "Learning image representations tied to ego-motion." Proceedings of the IEEE International Conference on Computer Vision. 2015.
>
> [2] Paccanaro, Alberto, and Geoffrey E. Hinton. "Learning distributed representations of concepts using linear relational embedding." IEEE Transactions on Knowledge and Data Engineering 13.2 (2001): 232-244.
>
> [3] Gao, Ruiqi, et al. "Learning grid cells as vector representation of self-position coupled with matrix representation of self- motion." Seventh International Conference on Learning Representations (2019).
>
> Q2: About the motivation. “Why do we care if the approach captures aspects of V1 for the tasks presented?” “Why do we need this over other methods that can better capture larger motions.”
>
> A2:  This is an important question.
>
> For tasks like optical flow estimation, current state of the art methods such as FlowNet2 use very complex deep neural networks, which are black box models. Our model is much simpler and is based on explicit vector and matrix representations. It is worthwhile to explore such models. Our new experiments also show that our method can achieve performances that are comparable to existing methods.
>
> Following your suggestion, we have strengthened the motivation of our work in the introduction.
>
> Your comment on evaluation is well taken. We have added evaluations on two more datasets in revision. One dataset is created in a similar manner as the public dataset of FlyingChairs. The other is the public MPI-Sintel dataset. See Subsections 5.1, 5.3 and Appendices E, F for details.
>
> About larger motions, the motions in the FlyingChairs dataset tend to be very big and abrupt, which does not really reflect typical motion behaviors observed in daily life. On the other hand, our model can be modified to a multi-resolution scheme to deal with larger motions. Currently we are exploring this direction.
>
> Q3: “‘Figure 1 illustrates the scheme of representation.’ Please provide more detail here on what is happening in the figure. The caption and reference here are not informative to what the figure is representing.”
>
> A3: Thanks for the suggestion. In the introduction of the revised version, we have included detailed explanation of Figure 1.
>
> Q4: “‘We obtain the training data by collecting static images for (It) and simulate the displacement field’. This is not self-supervised learning.”
>
> A4: Following your suggestion, we have removed the wording “self-supervised learning”, and changed the wording to “learning from image pairs with synthetic motions”.
>
> To be continued in the next message.

---

> > ### Author Response · Authors · 2019-11-15
> > **Response to Reviewer #5 (Part 2)**
> >
> >
> > Q5: “what guarantees are there that the model is learning to capture realistic motion behavior? Why not use adjacent video frames?”
> >
> > A5: Following your comment, we create another dataset called V1FlyingObjects, which separately applies affine transformations to the background scenes and foreground objects. For each training image pair, we jointly simulate the camera motion and the motion of the objects. This simulates more realistic motion behavior as you suggested.
> >
> > The V1FlyingObjects dataset consists of 14,411 image pairs. It is similar to the public dataset of FlyingChairs. The difference is that we use smaller motions and more types of objects. As mentioned above, the motions in FlyingChairs tend to be large and abrupt, which does not reflect typical motion behaviors. We shall release the V1FlyingObjects dataset to public.
> >
> > In addition, we have also tested the learned model on a public dataset, the MPI-Sintel. This is also a simulated dataset, but with special attention to realistic motions.
> >
> > Please see Subsections 5.1, 5.3 and Appendices E, F of the revised version for details of the new experiments.
> >
> > In supervised learning, our method, similar to other methods on optical flow estimation, requires ground truth displacements. Adjacent real video frames such as those in MUG usually do not have such ground truth information. We add description of MUG dataset in Subsection 5.1 to clarify the issue.
> >
> > Q6: About comparison with existing pre-trained model.
> >
> > A6: As we pointed out in Subsection 5.3, we did train state of the art models such as FlowNet2 on our dataset. But the trained model does not perform as well as the pre-trained model, possibly because our dataset is small. We thus reported the performance of the pre-trained model in order to be fair.
> >
> > Q7: About qualitative results on frame animation and frame interpolation.
> >
> > A7: Existing methods on optical flow are discriminative or predictive in nature, i.e., they take image pairs as input and output the optical flow estimation. Thus they cannot be used for frame animation and frame interpolation. Our model is a representational model or a generative model in some sense, in that we can generate the image frame given its vector representation. We use these qualitative experiments to illustrate this fact. We have moved the two Subsections to Appendices E and F.

---

### Official Review · AnonReviewer4 · 2019-11-03
**Official Blind Review #4**

**Rating:** 6

**Review:**

The hypothesis in this paper is that the primary purpose of the cells of the V1 cortex is to perceive motions and predict changes in the local image contents. The authors show that by learning from image pairs from a continuous sequence, both V1-like features and motion operators can be learned. I found the hypothesis and formulation reasonable, the numerical results are supportive, it's actually interesting to see that the proposed model's motion prediction outperforms the other dedicated models. Further, the authors used inference to infer the motion during learning, I think this is quite a novel topic to work on. Overall, this makes a good submission.

Here are some issues could be addressed further:

1. Section 3.3 introduces subvectors. This implicitly introduces an independence assumption when combined with a motion operator. Then in section 5, the authors studied the dimensionality of subvectors. If the subspaces are assumed to be 2, then this independence regularization is quite strong. This may not support the authors' claim that the prediction of motion is enough to achieve V1-like features and I tend to conclude the V1-like receptive fields come from the implicit independence constraint. I'd suggest an additional ablation experiment to verify the impact of the subspace assumption.

2. To model the motion,  we can directly use lie operators, the authors may want to discuss the connection between the suggested method and the Lie group approach.

3. I found some minor issues, e.g.:
       3.1 In Section 3.2 it's normalized tight frame (Parseval frame).
       3.2 In Equation 2 I understand it's a deconvolution, however, the notation is still not ideal.
       3.3 In the section paragraph of Section 3.2,  'the representation has the isometry property' and 'the vector representation also preserves the angle' should be switched?
       3.4 small typos like 'mortar cortex' -> 'motor cortex'.

**Experience Assessment:**

I have published one or two papers in this area.

**Review Assessment: Checking Correctness Of Derivations And Theory:**

I carefully checked the derivations and theory.

**Review Assessment: Checking Correctness Of Experiments:**

I assessed the sensibility of the experiments.

**Review Assessment: Thoroughness In Paper Reading:**

I read the paper at least twice and used my best judgement in assessing the paper.

---

> ### Author Response · Authors · 2019-11-15
> **Response to Reviewer #4**
>
> We are very grateful for your positive review and insightful comments.
>
> Q1: “I tend to conclude the V1-like receptive fields come from the implicit independence constraint.”
> A1: This is a deep insight that we agree. We have added a comment that this constraint is necessary for the emergence of V1-like receptive fields in Subsection 3.3.
>
> In Appendix G (Appendix D of the original version), we include an ablation study of subspace assumption. V1-like patterns also emerge when the dimensionality of subspace is higher (e.g., 4 or 6).
>
> Following your suggestion, we have added a result in Appendix E of the revised version, where we totally remove the assumption of sub-vectors. In this case, more blob-like patterns are learned.
>
> The sub-vectors may correspond to columns or modules of neurons, or capsules, i.e., neurons that form sub-groups.
>
> Q2: “connection between the suggested method and the Lie group approach.”
>
> A2: Thanks for the insightful suggestion. We have added a comment on this connection in Section 2.
>
> In our work, the displacements dx form a 2D Euclidean group. Our modeling of local motion dx is similar to the treatment of Lie group via Lie algebra by analyzing infinitesimal changes.
>
> The objects in the image may undergo more complex motions which form more complex Lie groups (e.g., rotations and translations). We can again represent the objects (e.g., their poses) by vectors, and represent the motions of the objects by matrices.
>
>
> We have followed your suggestions to correct those minor errors in the revision. Thank you for careful reading.

---

### Decision · Program_Chairs · 2019-12-19

**Decision:**

Reject

**Comment:**

The paper received mixed reviews. On one hand, there is interesting novelty in relation to biological vision systems. On the other hand, there are some serious experimental issues with the machine learning model. While reviewers initially raised concerns about the motivation of the work, the rebuttal addressed those concerns. However, concerns about experiments remained.